# Strategies Targeting Hemagglutinin as a Universal Influenza Vaccine

**DOI:** 10.3390/vaccines9030257

**Published:** 2021-03-13

**Authors:** Brianna L. Bullard, Eric A. Weaver

**Affiliations:** Nebraska Center for Virology, School of Biological Sciences, University of Nebraska–Lincoln, Lincoln, NE 68504, USA; bbullard@huskers.unl.edu

**Keywords:** universal vaccine, stalk, headless, chimeric, mosaic, consensus, centralized, epigraph, COBRA, ancestral

## Abstract

Influenza virus has significant viral diversity, both through antigenic drift and shift, which makes development of a vaccine challenging. Current influenza vaccines are updated yearly to include strains predicted to circulate in the upcoming influenza season, however this can lead to a mismatch which reduces vaccine efficacy. Several strategies targeting the most abundant and immunogenic surface protein of influenza, the hemagglutinin (HA) protein, have been explored. These strategies include stalk-directed, consensus-based, and computationally derived HA immunogens. In this review, we explore vaccine strategies which utilize novel antigen design of the HA protein to improve cross-reactive immunity for development of a universal influenza vaccine.

## 1. Introduction

Seasonal influenza epidemics infect between 10–15% of the global population each year [1]. Symptoms typically last between 5–15 days and include fever, headache, myalgia, and respiratory distress [2]. However, in at-risk patients such as the elderly and immunocompromised, influenza infection can result in severe morbidity and even mortality [3,4]. In addition to the substantial disease burden from seasonal influenza epidemics, there is a significant threat to global health from influenza pandemics. Highlighting this, the 2009 H1N1 swine influenza pandemic infected an estimated 24% of the global population [5].

Influenza virus is a negative-sense, single-stranded RNA viruses with a genome of 8 segments. These segments encode viral proteins including hemagglutinin (HA), neuraminidase (NA), nonstructural 1 (NS1), NS2, matrix 1 (M1), M2, nucleoprotein (NP), nuclear export protein (NEP), polymerase acid (PA), polymerase basic 1 (PB1) and PB2 [6]. There is substantial viral diversity in influenza virus, both through antigenic drift from the error-prone RNA polymerase and through antigenic shift from reassortment of the segmented viral genome resulting in novel reassorted viruses [6,7]. The major proteins on the surface of the virion are HA and NA. Influenza A viruses (IAVs) are classified based on 18 HA subtypes and 11 NA subtypes while influenza B viruses are classified into two lineages, Yamagata and Victoria. The 18 HA subtypes of IAV are divided into phylogenetic groups 1 and 2 (Figure 1). H1N1, H3N2, and both influenza B lineages currently circulate in the human population and cause seasonal epidemics. The H2N2 subtype caused a pandemic in humans in the 1957–1959 influenza seasons, but has, for the most part, been absent from the human population ever since [8,9]. Importantly, avian influenza strains H5, H7, and H9 subtypes have infected humans from zoonotic transmission but have shown limited transmission between humans [10]. However, the possibility of future mutations which enhance human-to-human transmission led the World Health Organization to recognize these subtypes as potential pandemic viruses [11,12,13].

Development of influenza vaccines is challenged by the substantial viral diversity of influenza virus [14]. The RNA polymerase of influenza virus has no proof-reading activity, which results in high mutation rates and substantial antigenic drift [15]. Therefore, current seasonal influenza vaccines are updated yearly and rely upon global surveillance to predict the future circulating seasonal strains [16]. The quadrivalent formulation contains an H1N1, H3N2, and two influenza B viruses, one from the Victoria and Yamagata lineage [2]. Current seasonal influenzas vaccines are effective at reducing morbidity and mortality from seasonal influenza infections [17], however vaccine effectiveness estimates range from 10–60% [16]. Vaccine efficacy relies on the correct prediction and close antigenic match between the chosen vaccine strain and the circulating influenza strain [16]. In addition, these seasonal vaccines are unlikely to provide protection from novel emergent pandemic strains (such as the 2009 H1N1 reassorted swine influenza virus).

Many strategies have been explored to increase the cross-reactivity of influenza vaccines, with the goal of developing a universal influenza vaccine. Importantly, HA is the predominant antigenic protein of influenza viruses and antibodies directed at HA are correlated with protection against influenza virus infection [18,19]. In this review, we explore vaccine strategies which target the HA protein for development of a universal influenza vaccine, with a particular focus on novel antigen design of HA to improve cross-reactive immunity.

## 2. Hemagglutinin Structure and Function

Hemagglutinin is the most abundant protein on the surface of influenza and functions in viral entry through receptor binding and membrane fusion [6]. HA is also the predominant antigenic protein and therefore shows the highest rates of adaptive evolution out of all the influenza proteins [20]. Although there are high levels of protein sequence diversity between the subtypes, the HA protein maintains required elements, such as the cleavage site, secretory signal, fusion domain, transmembrane domain, and cytoplasmic tail as well as common protein structural motifs [21].

HA is a glycoprotein which assembles as a homotrimer on the surface of the virion (Figure 2) [6]. Each monomer starts initially as a single polypeptide precursor (HA0) which is later cleaved into HA1 and HA2 subunits by host proteases. This cleavage is essential for maturation of the virus to an infectious virion [22]. The HA2 subunit is composed mostly of the stalk region of HA and the C-terminus which has a transmembrane domain with a cytoplasmic tail that anchors the HA protein to the envelope of the influenza virion. The HA1 subunit contains the signal peptide at the N-terminus and the globular head domain. This globular head domain contains the receptor binding site which binds sialic acid on the surface of the host cell and facilitates viral entry [7]. Upon internalization of the virion, acidification of the endosome induces a conformational change of HA, which exposes the N-terminus fusion peptide of the HA2 subunit. The fusion peptide then facilitates membrane fusion and release of the viral RNA into the cytoplasm of the host cell [22,23].

The globular head of influenza also contains the antigenic sites determined for both H1 and H3 [7]. Neutralizing antibodies for influenza are typically directed against these highly antigenic sites on the globular head and interfere with HA binding to sialic acid [7]. The HA protein of influenza virus has the ability to agglutinate red blood cells. Anti-influenza virus antibodies which bind to HA and inhibit the hemagglutination activity of HA are used as a surrogate measure of determining neutralizing antibody titers (HI titers) [18]. Human serological studies have demonstrated that HI titers of at least 1:40 correlate with protection from influenza infection [18,19]. Antibodies directed against the stalk domain of HA have different mechanisms of action, as discussed below, and cannot be measured using an HI assay.

## 3. Stalk-Directed Strategies

### 3.1. General Principles of Stalk-Directed Stratgies

One strategy to stimulate broadly reactive antibodies against the large diversity of HA is to target the more conserved stalk region. The stalk region of HA is occluded on the surface of the influenza virion and is therefore under less selective pressure from the immune system [24]. Consequently, the stalk region is more conversed as compared to the globular head of HA, although there is still protein sequence variability within subtypes, as measured by Shannon entropy (Figure 3). Bioinformatic analysis has shown that the stalk domain of HA is evolving at a slower rate than the head domain [25]. Protective efficacy mediated by head-directed antibodies primarily work through direct binding to HA in order to inhibit viral attachment to sialic acid on host cells, thereby preventing viral entry. In contrast, stalk-directed antibodies have been proposed to work through alternative mechanisms, such as inhibiting conformational changes at low pH to prevent virus release from the endosome and preventing maturation of virus by inhibiting cleavage of HA0 [26]. Additionally, indirect Fc-mediated functions have been reported, such as antibody dependent cell-mediated cytotoxicity (ADCC) and triggering of complement-dependent cytotoxicity (CDC; reviewed in [26]). This has been further supported by results indicating that stalk antibodies require Fc-mediated interactions for in vivo efficacy [27,28]. While antibodies directed to the head of HA are typically measured by HI titer, the non-classical effector functions of stalk-directed antibodies require alternative methods to measure antibody titers, such as ELISA or ADCC reporter assay.

Stalk-directed antibodies have shown cross-reactivity within subtypes and between multiple subtypes within a group and even between group 1 and 2 viruses. A comprehensive list of stalk-directed antibodies discovered have been previously described [29]. Researchers have explored many vaccination strategies to induce these stalk-directed antibodies (Figure 4), however induction of antibodies against the immunosubdominant stalk domain remains challenging in the presence of the immunodominant head domain. Efforts to overcome this challenge include development of “headless” HA constructs, hyperglycosylation of the head, expression of just the long alpha helix (LAH) domain of the stalk, and development of chimeric and mosaic HAs.

### 3.2. Design of Stalk-Directed Vaccines

#### 3.2.1. Headless HA Constructs

One strategy to increase antibodies directed to the immunosubdominant stalk region is through removal of the immunodominant head domain creating a “headless” HA. Importantly, HA is a metastable protein which undergoes extensive conformation changes at low pH during the infection cycle of influenza [35]. Removal of the HA1 head domain destabilizes the HA2 structure resulting in loss of antibodies targeting the native conformational epitopes. This was demonstrated in the first reported experiment developing a headless HA in 1983, which removed the HA1 domain of HA through chemical treatment with acid and a reducing agent [33]. However, this vaccine did not show protective efficacy, likely due to denaturation of conformational stalk epitopes. Other efforts to express only the HA2 subunit in systems such as recombinant baculovirus or *E. coli* have resulted in stalk antigens which lack the native confirmation and are not recognized by neutralizing anti-stalk antibodies [34,35,36].

Multiple strategies have since been explored to express the HA stalk region in a native-like, neutral-pH conformation. One group stabilized the stalk through inserted specific mutations intended to destabilize the low pH conformation of HA2 thereby pushing the protein to a neutral pH conformation. This approach has been applied to both H1 and H3 proteins [37,38]. The stabilized HA2 protein was expressed in *E. coli* and folded into a neutral-pH conformation. However, mice vaccinated twice with the stabilized H1 stalk protein were protected from mortality, but not morbidity (~18% weight loss), after challenge with a lethal homologous strain [38]. Another group stabilized the H1 or H3 HA2 domain through inclusion of stabilizing linker peptides and vaccinated mice with two doses of DNA protein expression plasmids followed by a virus-like particle (VLP) formulation [65]. Vaccination with this headless HA completely protected mice from a challenge with a homologous virus strain (~5% weight loss) and induced greater in vitro heterosubtypic cross-reactive antibodies.

Other groups have aimed to express stable headless HA in a trimer conformation either as a soluble protein [39,40,41,42,43] or on the surface of virus-like particles (VLP) [44] or nanoparticles [45,46,47]. One group developed a soluble “mini-HA” H1 stalk trimer through multiple structure-based mutations [39]. Three immunizations with this “mini-HA” H1 stalk completely protected mice from weight loss and death after lethal challenge with either heterologous H1 or heterosubtypic H5 influenza virus. Sera from these mice had both neutralizing and ADCC effector functions. Another group stabilized an H1 stalk trimer through six iterative cycles of structure-based mutations and displayed the stalk on the surface of a nanoparticle [45]. Three immunizations with these nanoparticles completely protected mice from heterosubtypic challenge with H5 influenza virus but showed only partial protection in ferrets. Vaccinated mice and ferrets showed strong in vitro antibody binding against group 1 subtypes H1, H2, H5, and H9 with some weak responses to group 2 subtypes H3 and H7. However, there was limited neutralizing antibodies detected against heterosubtypic strains, indicating that protection is likely mediated by other stalk antibody-dependent mechanisms, such as ADCC or CDC. This stabilized H1 trimer nanoparticle vaccine has since progressed into a phase I clinical trial with 52 participants and is expected to conclude December 2021 (NCT03814720).

#### 3.2.2. Chimeric HA

To overcome the instability of headless stalk constructs while still boosting stalk-directed antibodies, a chimeric HA protein prime/boost strategy was developed. In this strategy, multiple sequential immunizations of chimeric HA proteins containing the same stalk region, but ‘exotic’ HA heads, results in a boosting of stalk-directed immunity. A major benefit of this approach is that the full-length HA is expressed, thereby presenting the stalk domain in the correct conformation. This chimeric strategy has been explored for use as H1 [51,52,53,54,55,56,57,58], H3 [59], and influenza B virus vaccines [60].

To boost stalk immunity against H1, mice were sequentially vaccinated with three doses of chimeric HA which all had the same H1 stalk but head regions from H9, H6, or H5 [52]. Mice were completely protected from lethal challenge with three homosubtypic H1 viruses. The authors also explored heterosubtypic protection and, to rule out the contribution of head-directed antibodies, immunized mice with a similar chimeric prime boost strategy but replaced the head domain of the corresponding challenge strain with a H1 head instead. Vaccination with this strategy completely protected mice from death after lethal heterosubtypic challenge with H5, H6, and H3 viruses, however weight loss data is not reported, although the authors state that only minimal weight loss was observed. Efficacy of this vaccine was demonstrated in a preclinical ferret model, where vaccination with the chimeric prime/boost strategy reduced viral nasal wash titers after challenge with a heterologous H1 virus [51] and heterosubtypic H6 virus [57], and demonstrated durability of protection against homologous H1 challenge up to 1 year after immunization [58]. Delivery of the H1 chimeric antigens has been explored utilizing multiple vaccine platforms, including a DNA prime with recombinant protein boosts [52], recombinant live-attenuated virus and inactivated virus [55,56,57], vesicular stomatitis virus (VSV) viral vectors [51,53], and adenovirus vectors [51].

Importantly, results from a phase I clinical trial of 65 participants have been reported [61,62]. Participants received two immunizations with different combinations of chimeric HA live-attenuated virus, inactivated virus, or inactivated virus plus adjuvant vaccines. Vaccination was found to be safe and to successfully induce anti-stalk H1 antibodies. The best results were observed in participants who received two doses of chimeric HA inactivated adjuvanted virus, in which anti-stalk antibodies persisted at ~4-fold above baseline for up to 1.5 years. Mice who received a passive transfer of sera from vaccinated participants showed a trend towards reduced weight loss as compared to mice who received sera from the placebo group after challenge with recombinant H1 stalk virus. This clinical trial supports the ability of this chimeric HA vaccine strategy to induce anti-stalk antibodies in humans, however clinical trials are needed to demonstrate the efficacy of these stalk-directed antibodies in protecting humans from infection. While the most progress has been made using the H1 subtype vaccine, this strategy has also been explored for H3 [59] and influenza B [60] with promising results. Combining these candidate immunogens into a multivalent vaccine has not yet been explored.

#### 3.2.3. Hyperglycosylated HA

Another strategy to increase stalk-specific responses is to hyperglycosylate the HA1 head region in order to ‘mask’ the immunodominant epitopes, thereby directing the response to the stalk [30]. Three immunizations of mice with a hyperglycosylated H1 protein induced stronger anti-stalk antibodies against the homologous H1 stalks than a wild type HA [32].The hyperglycosylated H1 also induced greater cross-reactive antibodies to two heterologous H1 viruses and a heterosubtypic H5 virus. Hyperglycosylated H1 vaccinated mice were protected from mortality, but not morbidity (~10% weight loss), after challenge of mice with a chimeric H1 stalk virus.

This strategy has also been applied to target the H5 stalk [30,31]. Immunization of mice with a replication-defective adenovirus vector expressing a hyperglycosylated H5 clade 1 HA and boosted with recombinant protein induced greater cross-reactive antibodies against three heterologous clade 2 viruses as compared to a wildtype virus [31]. While this vaccination induced homosubtypic antibodies against H5, it did not result in significant antibody responses to other group 1 viruses (H1, H3, H9). Vaccination with hyperglycosylated H5 protected mice from mortality, but not morbidity (~13% weight loss), after challenge with a heterologous clade 2 virus, with no significant difference in weight loss observed between the hyperglycosylated H5 or wildtype HA vaccine. Therefore, while hyperglycosylation of the HA1 head does appear to increase anti-stalk antibodies as compared to the wildtype HA, vaccination does not protect mice from severe influenza morbidity, even against a homosubtypic challenge.

#### 3.2.4. “Mosaic” HA

Another strategy to silence the immunodominance of the HA1 head subunit and direct the immunity to the stalk region involves substitution of the immunodominant antigenic sites on the HA1 head, yielding a “mosaic” HA1 region. One strategy silenced the immunodominant antigenic sites of the H3 head by replacing them with amino acid sequences from an avian HA protein (H10 or H14) [63]. Mice vaccinated with this recombinant mosaic H3 protein induced more H3 stalk-directed antibodies than the commercial inactivated seasonal vaccine. Passive transfer of sera from mice vaccinated with this adjuvanted mosaic recombinant H3 protein protected mice from death after lethal challenge with two heterologous H3 viruses, however mice still experienced severe weight loss (~20% weight loss).

This strategy was also applied to an influenza B vaccine, where immunodominant antigenic sites of an influenza B Yamagata lineage virus was replaced with amino acid sequences from H5, H8, H11, or H13 [64]. Vaccination of mice with a DNA prime and two recombinant protein boosts of this mosaic head protein resulted in complete protection from weight loss and death after challenge with a lethal Yamagata and Victoria strain. Antibodies induced after vaccination with the mosaic HA showed cross-reactive ELISA antibodies against three Yamagata lineage viruses and three Victoria lineage viruses. No HI antibodies or neutralizing antibodies were detected, but there were antibodies detected through an ADCC reporter assay, indicating that efficacy of this mosaic vaccine was conferred primarily through Fc-mediated effector functions.

#### 3.2.5. LAH Fragment

In an attempt to overcome the challenges with expression of a stable headless stalk and the immunodominance of the HA1 head, some strategies instead target only a small portion of the stalk region, such as the long alpha helix (LAH) or fusion peptide. One strategy conjugated the LAH of H3 to a carrier protein [48]. Two immunizations of mice with this LAH vaccine showed complete protection from death after a lethal challenge with heterologous H3 virus, however mice still exhibited severe weight loss (~15% weight loss). This LAH vaccine only partially protected from death after lethal heterosubtypic H5 challenge and did not protect against lethal heterosubtypic H1 challenge. There was similar findings by another group, which expressed the LAH and fusion peptide from H5 in *E. coli* and refolded the protein from inclusion bodies [50]. Mice immunized with this protein were completely protected from weight loss and death after challenge with the homologous H5 virus but only partially protected from death after challenge with a heterologous H5 virus or a heterosubtypic H1 virus. Additionally, vaccination of mice with the LAH of H1 expressed on a hepatitis B virus-like particle (VLP) demonstrated complete protection from mortality after challenge with a lethal homologous H1 virus, heterologous H1 virus, and heterosubtypic H9 virus, although mice still showed severe weight loss of ~15%, 7%, and 19%, respectively [49]. Therefore, although vaccination with the LAH does increase heterosubtypic protection, the morbidity after challenge is often still severe.

### 3.3. Challenges Facing Stalk-Directed Strategies

Although these stalk-directed vaccination strategies do induce greater cross-reactive immunity, the weak immunogenic nature of the stalk domain necessitates as many as three immunizations to induce this immunity. In addition, animals still exhibit severe morbidity even after challenge with homosubtypic strains, often losing more than 10% of their initial starting weight, although they are protected from mortality. This is likely due to the Fc-mediated mechanism of action found to be a large contributor to protection in many stalk-directed strategies, as these mechanisms are not neutralizing but rather contribute to viral clearance after infection [26]. This indicates that although the breadth of reactivity is increased with these stalk-directed approaches, the robustness of the protection is diminished [66].

Additionally, although the stalk region of HA is more conserved than the head region, there is still variability within subtypes (Figure 4) and the stalk region is still susceptible to selective pressures from the immune system [67]. This is highlighted by the discovery that pre-existing stalk-directed antibody titers select for a stalk-antibody escape mutant after human influenza challenge [67]. Furthermore, while these stalk-directed antibodies do confer protection from mortality in animal models, there is still conflicting literature about the efficacy of these anti-stalk antibodies in humans. One study utilizing a human household transmission experimental design found a correlation between HA stalk-directed antibodies and protection from infection [68]. In contrast, another study using human samples demonstrated that there was no significant correlation between stalk antibodies and protection from influenza virus infection after adjustment for head-specific antibodies [69]. In addition, a phase II trial examining the therapeutic efficacy of a monoclonal stalk-directed antibody showed no clinically significant effect on influenza disease [70]. These conflicting results draw into question the protective efficacy of these anti-stalk antibodies in humans.

## 4. Consensus-Based Strategies

### 4.1. General Principles of Consensus-Based Stratgies

Unlike stalk-directed strategies, consensus-based approaches target the full-length HA protein and aim to design an HA protein which is broadly representative of the diverse HA population. These HA genes are computationally designed and therefore are synthetic in nature. Although the central concept behind consensus-based strategies is similar, many different approaches have been developed (Figure 5). Importantly, unlike stalk-directed strategies, consensus-based approaches typically induce HI antibodies directed against the head region of HA, which is generally accepted as a correlate of protection against influenza infection in humans [29,71]. Consensus-based approaches are typically developed for a single subtype of influenza, with the intention of expanding the approach to multiple subtypes and combining the vaccine constructs into a single multivalent vaccine.

### 4.2. Design of Consensus-Based Vaccines

#### 4.2.1. Consensus Hemagglutinins

Consensus HA genes are constructed by taking the most common amino acid at each position of the HA after aligning a protein sequence population. A consensus strategy has been explored by multiple groups to target highly pathogenic avian influenza (HPAI) H5N1 virus [72,73,74,75], which poses a substantial pandemic threat. In addition, H5N1 has high viral diversity, with multiple clades and subclades [76]. Two immunizations of mice with a consensus H5 gene expressed in a DNA plasmid induced cross-reactive antibodies against multiple different H5 viruses from two clades [72]. Vaccination conferred complete protection from morbidity and mortality after challenge with two H5 clades but only partial protection after challenge with more distantly related H5 viruses. Additionally, a consensus H5 protein expressed in VLPs completely protected chickens from lethal challenge with two H5 viruses from separate clades [74]. Importantly, these consensus immunogens are designed using all H5 protein sequences in the database, which can lead to geographical sampling/sequence bias and result in a consensus HA that might not accurately reflect the HA diversity of the population.

#### 4.2.2. Micro-Consensus Hemagglutinin

While other consensus designs develop a single consensus HA per subtype, Elliott et al. (2018) explored the efficacy of four micro-consensus immunogens to improve cross-reactivity to the H3 subtype [77]. These four micro-consensus genes were expressed in a DNA plasmid and administered twice as cocktail. Vaccination of mice with this cocktail induced strong antibody responses against 8 strains circulating between 1968 and 2014 as measured by ELISA and induced significant HA-specific cellular immunity. Vaccination also protected mice from lethal challenge with two H3N2 strains, with 5–10% weight loss. Importantly, this study finds that a cocktail of consensus immunogens might improve cross-reactivity of highly diverse HA populations, such as the H3 subtype.

#### 4.2.3. Centralized Hemagglutinins

Another consensus-based strategy aims to develop a synthetic HA which localizes to the central node of the phylogenetic tree, thereby minimizing genetic and antigenic differences of unmatched strains. An important limitation to consensus strategy described above is the threat of sampling bias leading to generation of a synthetic HA gene which does not accurately represent the diversity of the population and is biased towards an overrepresented geographical location. Weaver et al. (2011) overcame this by developing a centralized H1 gene using selected representative wildtype HA protein sequences from each major branch of the phylogenetic tree and designing a consensus of those sequences [78]. In this way, each branch is equally represented to prevent sampling/sequencing bias and, as a result, this synthetic HA gene localizes to the central node of the tree. This centralized H1 gene was expressed in a replication-defective adenovirus vector and vaccination of mice resulted in better cross-protection against three H1 strains as compared to mismatched wild type HAs or traditional influenza vaccines after lethal influenza challenge [78]. This strategy was then applied to develop H3 and H5 centralized genes which also demonstrated improved homosubtypic cross-protection [79]. Importantly, a combination of H1, H2, H3, and H5 centralized genes into a single multivalent vaccine did not diminish the increased cross-reactive protection [80]. A single immunization at the high dose of this multivalent formulation demonstrated complete protection from morbidity and mortality after lethal challenge with three H1 strains, three H5 strains, and one H3 strain with partial cross-protection against another two H3 strains. This validates the approach of designing multiple subtype-specific broadly reactive HA immunogens which can then be combined into one multivalent vaccine without reduction in cross-reactivity or interference between immunogens.

#### 4.2.4. COBRA Hemagglutinins

Like the centralized HA design, another antigen design method called computationally optimized broadly cross-reactive antigen (COBRA) aims to minimize sampling/sequencing bias in the target population. The COBRA strategy achieves this through multiple rounds of consensus generation. A COBRA vaccine strategy has been developed and tested for H1 [81,82,83,84,85], H2 [86], H3 [87,88], H5 [89,90,91,92,93,94,95,96], and swine H1 [97]. These COBRA immunogens have primarily been expressed using a VLP platform but have also been explored using ferritin nanoparticles [84] and recombinant live influenza virus [83]. This strategy was first explored as a H5 vaccine targeting only clade 2 and was tested in mice, ferrets [89,90], chickens [95], and non-human primates (NHP) [91]. Vaccination of NHP induced cross-reactive HI antibody titers against multiple clade 2 viruses and against a clade 1 and 7 influenza virus. This strategy was then expanded to target the entirety of the H5 diversity by using a cocktail of three COBRA immunogens designed for human clade 2, human and avian clade 2, and all H5 clades [92]. Vaccination of mice with two doses of this cocktail induced protective HI titers to twenty-five H5 viruses from eleven H5 clades/subclades, however cross-protection was only evaluated after immunization with individual COBRA immunogens and was not evaluated after vaccination with the cocktail.

The COBRA strategy was also evaluated for protection against seasonal and pandemic H1 strains [82]. Nine different COBRA H1 immunogens were designed and four immunogens were selected for further investigation based on their increased cross-reactivate HI titer and protection. These immunogens were evaluated in sequential prime/boost immunizations strategies or as a heterologous cocktail. Results found that the broadest HI titers were induced by a combination of COBRA immunogens designed using both seasonal and pandemic H1 strains, with a trivalent cocktail demonstrating protective HI titers to ten of the fifteen H1 strains [82]. This increased HI cross-reactivity extended to a pre-immune ferret model [81]. However, cross-protection was only evaluated against a single pandemic strain.

A similar strategy was used for the H3 subtype, in which seventeen CORBA immunogens were designed with only four of these immunogens showing promising HI titers in mice [87]. The four immunogenic H3 COBRA immunogens were further explored in a ferret model [88]. Although vaccination of naïve mice showed promising cross-reactivity, vaccination of naïve ferrets showed low HI titers and narrow cross-reactivity, with the most cross-reactive immunogen showing protective HI titers to only six of the thirteen H3 strains. In contrast, ferrets that were pre-immune to a historical H3 strain prior to vaccination showed increased cross-reactive HI titers against the entire panel of 13 viruses after vaccination with COBRA immunogens as compared to vaccination with a wildtype HA. This led the authors theorize that, although there was limited efficacy in naïve ferret, the COBRA immunogens were more effective than wildtype HA at recalling broad cross-reactive memory B cells from previous influenza infection. Therefore, while this H3 COBRA vaccine might boost cross-reactive immunity in pre-immune adults, more research needs to be performed to examine potential efficacy in naïve populations, such as children. In addition, although the COBRA strategy has been explored for multiple influenza subtypes individually, COBRA has not yet been explored as a multivalent formulation targeting multiple subtypes.

### 4.3. Challenges Facing Consensus-Based Approaches

Consensus-based approaches target the full-length HA protein and therefore typically induce antibodies directed against the variable head region. For this reason, these vaccines are subtype-specific and do not show the cross-reactivity between subtypes of the same phylogenetic group, as is observed in stalk-directed strategies discussed above. However, antibodies induced by these consensus-based approaches have improved cross-reactivity within a subtype and often demonstrate HI activity which is a correlate of protection against influenza infection in humans [29,71]. Therefore, what these approaches lack in breadth of immunity between subtypes, they make up for in robustness of protection within subtypes.

Consensus-based influenza vaccines based on the HA protein are a more recent concept than the stalk-directed strategies and therefore have not progressed to human clinical trials yet. In addition, unlike stalk-based strategies, little work has been performed to evaluate the potential for escape mutants from pre-existing immunity induced by consensus vaccines. The subtype-specific immunity induced by consensus vaccines will necessitate a multivalent cocktail of HA immunogens to protect against the multiple subtypes currently circulating in humans. Further work to demonstrate that a multivalent vaccine containing many immunodominant antigenic sites from different subtypes does not reduce efficacy through interference. However, the centralized HA approach has shown that vaccination of mice with a quadrivalent vaccine showed no reduction in efficacy, indicating that this a promising strategy to induce a robust protective immune response against multiple subtypes relevant to human health.

## 5. Computational Algorithms for Immunogen Design

### 5.1. General Prinicples of Computational Algorithm Approaches

Similar to consensus-based strategies, approaches using computational algorithms target the full-length HA protein and aim to develop a synthetic HA which is broadly representative of the diverse influenza virus population. However, these strategies use complex computational algorithms in order to logically design a broadly reactive HA immunogen in silico.

### 5.2. Design of Computional Algorithm Vaccines

#### 5.2.1. Ancestral Hemagglutinin

A phylogenetic algorithm called ANCESCON reconstructs the ancestral HA gene using marginal and joint reconstruction methods [98]. Ducatez et al. (2011) used this algorithm to develop a cross-reactive H5 vaccine, in which the putative ancestral gene for avian H5 was predicted and expressed in a recombinant, replication-competent influenza virus [76]. Vaccination of ferrets with the ancestral gene induced in vitro cross-reactivity to five H5 clades and protected ferrets from death after lethal challenge with three H5 viruses from two clades. However, a wild type H5 reference virus also showed similar levels of cross-reactive immunity and protection. Crucially, this study demonstrated a proof-of-concept that synthetic HA genes could be expressed using an inactivated recombinant virus which is a currently licensed influenza vaccine platform.

This ANCESON algorithm was also used to develop an ancestral avian H9 immunogen that was expressed in a Modified Vaccinia Ankara (MVA) vector [99]. Intramuscular vaccination of chickens resulted in poor cross-reactive HI titers, with measurable titers against only one of the ten avian influenza virus strains and did not reduce viral shedding after challenge. Therefore, although reconstruction of the ancestral HA gene aids in our understanding of the evolutionary processes underlying the HA protein, the improved cross-reactive efficacy of these immunogens has yet to be demonstrated. In addition, phylogenetic uncertainty from long branches and genetic drift might weaken this approach [100].

#### 5.2.2. Mosaic Algorithm

Another algorithm, called the Mosaic Vaccine Designer tool, aims to maximize the potential epitope coverage of the target viral population [101,102]. A graphic visualization of the algorithm approach has been previously described [102]. Simplified, full-length HA protein sequences undergo repeated rounds of in silico recombination and the recombinant HA protein which has the best epitope coverage of the original target population is selected as a single mosaic HA immunogen. Crucially, this algorithm results in a mosaic immunogen that typically has better epitope coverage than a consensus strategy. Mosaic HA immunogens have been designed for H5 [103,104,105,106] and H1 [107]. A mosaic H5 expressed in MVA completely protected mice from morbidity and mortality after lethal virus challenges with four H5 viruses from three clades and provided protection from mortality, but not morbidity (~10% weight loss), after challenge with a heterosubtypic H1 virus [103]. T-cell depletion and passive transfer studies found that homosubtypic protection is primarily from cross-reactive antibodies, while the heterosubtypic protection to H1 is from cross-reactive CD8+ cells [104]. Vaccination with the mosaic H5 immunogens was also shown to increase early viral clearance in the lungs of NHP after challenge with H1 virus [105] and eliminate viral shedding in chicks after challenge with H5 virus [106]. This mosaic algorithm was also used to design a H1 mosaic immunogen which was expressed in a replication-defective adenovirus vector [107]. Vaccination of mice showed increased cross-reactive ELISA antibodies and T-cell responses to a panel of four H1 viruses as compared to mismatched wild type HA immunogens. In addition, the high dose immunization of mosaic H1 provided complete protection from morbidity and mortality after challenge with three H1 viruses. These studies demonstrate the efficacy of a computational algorithm which creates a vaccine antigen optimized to include the most common epitopes from the target population.

#### 5.2.3. Epigraph Algorithm

Similar to the mosaic algorithm, the epigraph algorithm aims to design vaccine antigens which contain the most common epitopes from the target population, thereby representing the diversity of HA protein sequences and biasing towards antigen recognition by the immune system [108,109]. The epigraph algorithm uses a faster graph-based approach and creates a user-defined cocktail of computationally designed HA proteins. A graphic visualization of the algorithm approach has been previously described [108]. The epigraph algorithm has been used to design a swine H3 immunogen, with the goal of reducing the pandemic potential of H3 from zoonotic transmission events [110]. This epigraph HA was expressed in a replication-defective Adenovirus vector and vaccination of mice induced increased cross-reactive antibodies as compared a wildtype HA or the commercial vaccine, with protective HI titers to fourteen of the twenty diverse swine H3 influenza virus strains. Importantly, this improved cross-reactive antibody response induced by the swine H3 Epigraph immunogens was also seen in swine, the target animal of this vaccine. Of interest, this swine H3 epigraph vaccine also induced cross-reactive antibodies with human H3 strains, suggesting the possibility of reduced reverse zoonotic events. Challenge studies in mice showed greater cross-protective efficacy after challenge with three swine H3 strains, however challenge studies in swine are needed to demonstrate this protective efficacy in the target animal. The broad cross-reactive immunity induced by this vaccine demonstrated the efficacy of a cocktail of computationally designed HA immunogens.

### 5.3. Challenges Facing Computational Algorithm Approaches

Computational algorithm approaches face many similar challenges as the consensus-based approaches, such as the possibility of escape mutants and subtype specific immunity induced by vaccination. Computational power continues to progress, and consequently computational algorithm approaches are the most recent advances in novel antigen design for influenza vaccines. Therefore, these approaches are still in their infancy and further research is needed to demonstrate their efficacy. However, the promising results discussed above lend support towards a computational algorithm improving cross-reactive immunity against the HA protein of influenza.

## 6. Perspectives

In this review we have explored multiple novel immunogen design strategies which aim to improve cross-reactive immunity against the HA protein for the development a universal influenza vaccine. These strategies target either the more conserved stalk region of HA or aim to design a synthetic full-length HA which is representative of the influenza viral diversity, either through a consensus-based approach or computational algorithms. Some of these strategies have been explored in depth, with headless stalk constructs and chimeric HA progressing to human clinical trials. Other strategies are more recently developed but still present a promising approach towards a universal influenza vaccine. Importantly, the strategies presented here all utilize different vaccine platforms, from viral vectors, to purified recombinant protein, to VLPs. The vaccine platform and immunization strategy can have a profound effect on the cross-reactive immunity induced by vaccination, which therefore makes these strategies difficult to compare head-to-head. However, in general, stalk-directed strategies induce antibodies which cross-react with multiple subtypes within a phylogenetic group and protect from mortality, but still result in significant morbidity after challenge. In contrast, full-length synthetic HA vaccines induce only subtype-specific immunity, but protect mice from both morbidity and mortality. This is likely due to the different mechanisms of action of stalk-directed versus head-directed antibodies. Therefore, a tradeoff is made in which increased cross-reactivity leads to a reduction in the robustness of protection. Overall, many of these novel antigen design strategies have shown promise in improving the cross-reactive immunity to influenza and should be further explored as a universal influenza vaccine.

## Figures and Tables

**Figure 1 vaccines-09-00257-f001:**
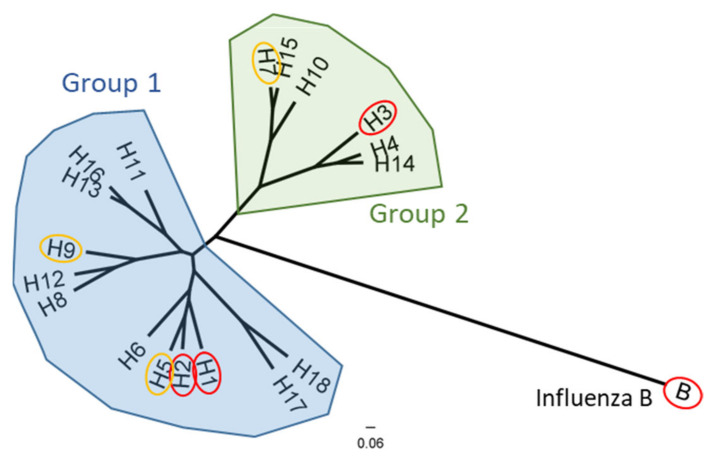
Phylogenetic relationship of the influenza A and influenza B hemagglutinin proteins. Group 1 IAVs (blue) and group 2 IAVs (green) phylogenic groups are indicated. Subtypes which are known to circulate in humans (or have previously like H2) are circled in red, while subtypes which infect avian species but are recognized to have significant pandemic potential for zoonotic transmission are circled in yellow. Representative strains for each HA subtype were aligned using ClustalW alignment and a maximum-likelihood phylogenetic tree was constructed using PhyML3.3 on Geneious 11.1.5.

**Figure 2 vaccines-09-00257-f002:**
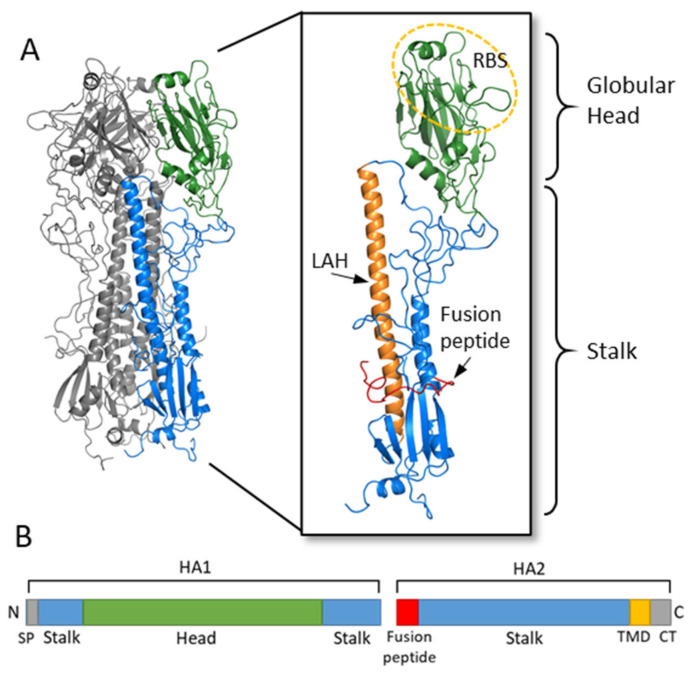
Hemagglutinin structure and functional regions. (**A**) The HA trimer of an H3N2 virus was downloaded from the Protein Data Bank (PDB: 1HGF; A/X-31) and visualized with PyMOL. Two monomers are colored in grey while the third monomer shows the head region in green and the stalk region in blue. An enlarged view of the HA monomer is further colored to show the fusion peptide in red, the long alpha helix (LAH) in orange, and the receptor binding site (RBS) on the head circled in yellow. (**B**) A linear schematic of the HA molecular is shown below. The head domain (green) is on the HA1 subunit, while the stalk domain (blue) spans the C- and N-terminus of HA1 along with most of HA2. At the N-terminus of HA1 is the signal peptide (SP) while at the N-terminus of HA2 is the fusion peptide. The transmembrane domain (TMD) and cytoplasmic tail (CT) are at the C-terminus of HA2.

**Figure 3 vaccines-09-00257-f003:**
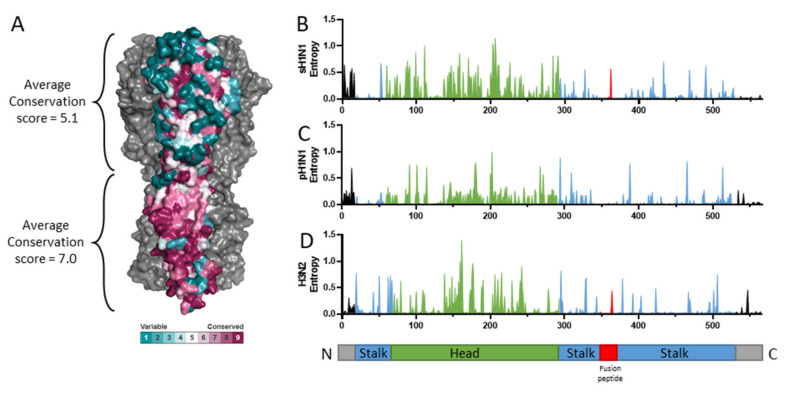
Conservation of the stalk of hemagglutinin. (**A**) The conservation scores of the pre-pandemic seasonal H1N1 (sH1N1) influenza viruses were calculated using the ConSurf server (https://consurf.tau.ac.il/ (accessed on 13 March 2021)) and visualized using PyMOL. Protein sequence variability was determined with the Shannon entropy server (https://www.hiv.lanl.gov/content/sequence/ENTROPY/entropy_one.html (accessed on 13 March 2021)) using ClustalW aligned HA protein sequences from pre-pandemic seasonal H1N1 ((**B**); sH1N1), pandemic H1N1 ((**C**); pH1N1), and H3N2 (**D**). All human isolates with complete HA protein sequences (duplicates and lab strains excluded) were downloaded from the Influenza Research Database, resulting in 962 strains for sH1N1 (strains up to 2009), 7423 strains for pH1N1 (strains during and after the 2009 pandemic), and 9043 strains for H3N2. The stalk, head, and fusion peptide are colored in blue, green, and red, respectively.

**Figure 4 vaccines-09-00257-f004:**
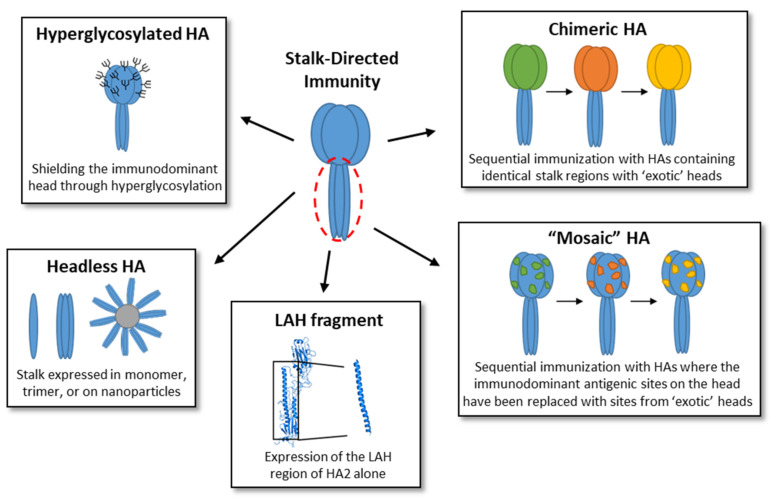
Vaccine strategies to increase stalk-directed immunity. Strategies used to induce stalk-directed immunity, such as hyperglycosylation of the HA [30,31,32], development of a headless HA [33,34,35,36,37,38,39,40,41,42,43,44,45,46,47], expression of the LAH fragment alone [48,49,50], and chimeric [51,52,53,54,55,56,57,58,59,60,61,62] and mosaic [63,64] HA proteins, have been illustrated and briefly described here.

**Figure 5 vaccines-09-00257-f005:**
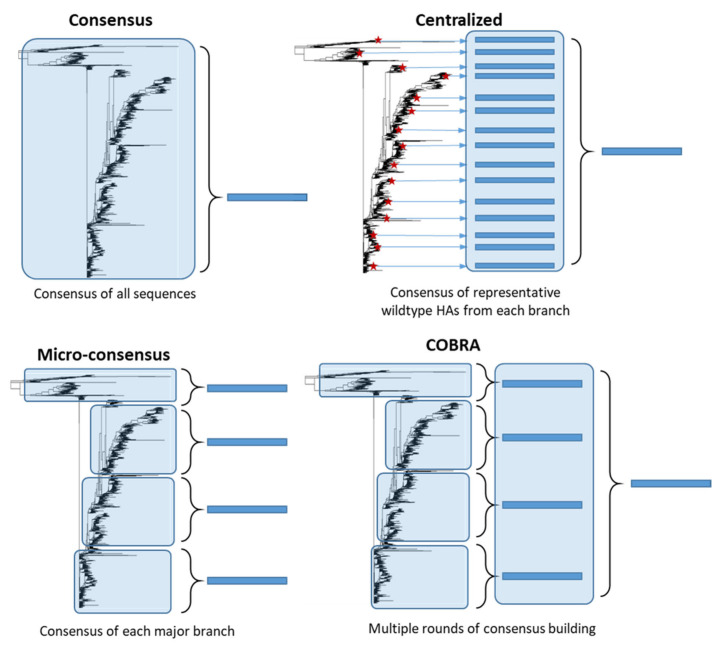
Consensus-based antigen design strategies. Consensus strategies are designed by aligning the target HA protein sequence population and determining the most common amino acid at each position. ‘Consensus’ immunogens are designed using all the HA protein sequences from the phylogenetic tree. ‘Micro-consensus’ immunogens are created by taking a consensus of each major branch of the phylogenetic tree and delivering the immunogens as a cocktail. ‘Centralized’ immunogens aim to reduce sampling bias by taking a representative wildtype HA (red stars) from each branch of the phylogenetic tree and designing a consensus of those HA protein sequences. ‘COBRA’ immunogens also aim to reduce sampling bias and use multiple rounds of consensus building.

## Data Availability

No data was generated in this study. All review material was accurately referenced.

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
