# Peer review of "Strategies Targeting Hemagglutinin as a Universal Influenza Vaccine"

_vaccines, 2021, doi:10.3390/vaccines9030257_

Round 1

Reviewer 1 Report

Thank you for the opportunity to review this manuscript. Bullard et al summarized the recent development and knowledge about a universal vaccine strategy against Influenza viruses. This manuscript is well-written and structured, therefore I learned a lot from this review manuscript. There are a few suggestions that might improve this review manuscript. Please see my specific comments stated in the comments for authors.

  1. I would like to suggest you add some more detailed description of why the development of effective vaccine systems against RNA viruses such as Influenza is so challenging (e.g., comparing mutation rates).
  2. Throughout the text, I confused the word "sequence". You mean "nucleotide sequence"? or "amino acid sequence"?. Please describe this accurately.

Author Response

Dear Reviewers,

Thank you for your thoughtful suggestions. Please see the responses to your comments below. You will find that we corrected the manuscript and included the inserted text in the response in order to help you identify improvements. We thank the reviewers for their time and effort. 

Sincerely,

Eric Weaver

Reviewer 1

Thank you for the opportunity to review this manuscript. Bullard et al summarized the recent development and knowledge about a universal vaccine strategy against Influenza viruses. This manuscript is well-written and structured, therefore I learned a lot from this review manuscript. There are a few suggestions that might improve this review manuscript. Please see my specific comments stated in the comments for authors.

I would like to suggest you add some more detailed description of why the development of effective vaccine systems against RNA viruses such as Influenza is so challenging (e.g., comparing mutation rates).

We have added to the introduction to include the high levels of mutation rates of RNA viruses. The intro now reads: “Development of influenza vaccines is challenged by the substantial viral diversity of influenza virus [14]. The RNA polymerase of influenza virus has no proof-reading activity, which results in high mutation rates and substantial antigenic drift [15]. Therefore, current seasonal influenza vaccines are updated yearly and rely upon global surveillance to predict the future circulating seasonal strains [16].

Throughout the text, I confused the word "sequence". You mean "nucleotide sequence"? or "amino acid sequence"?. Please describe this accurately.

We have ensured that every occurrence of the word ‘sequence’ is defined as a HA protein sequence or amino acid sequence.

Reviewer 2 Report

The authors have reviewed the strategies for generating a universal influenza vaccine by targeting the hemagglutinin protein. The review is clearly organized and comprehensively written and covers a broad range of complementary strategies.

Minor points:

  • Lines 97 to 99: this sentence should be rephrased.
  • Figure 2: The “RBS” acronym should be added in the legend after region binding site (line 108). Dividing the figure in 2 panels for the 3D and linear views respectively would maybe make it clearer.
  • Figure 4 and/or associated text: references should be provided for each strategy.
  • Lines 177, 250 and following occurrence: specify that the evaluation of morbidity is by weight loss (not just loss).

Author Response

Dear Reviewers,

Thank you for your thoughtful suggestions. Please see the responses to your comments below. You will find that we corrected the manuscript and included the inserted text in the response in order to help you identify improvements. We thank the reviewers for their time and effort. 

Sincerely,

Eric Weaver

Reviewer 2

The authors have reviewed the strategies for generating a universal influenza vaccine by targeting the hemagglutinin protein. The review is clearly organized and comprehensively written and covers a broad range of complementary strategies.

Minor points:

Lines 97 to 99: this sentence should be rephrased.

We have rephrased this sentence to read: “The HA protein of influenza has the ability to agglutinate red blood cells. Anti-influenza antibodies which bind to HA and inhibit the hemagglutination activity of HA are used as a surrogate measure of determining neutralizing antibody titers (HI titers).”

Figure 2: The “RBS” acronym should be added in the legend after region binding site (line 108). Dividing the figure in 2 panels for the 3D and linear views respectively would maybe make it clearer.

We have added RBS to the legend and also split Figure 2 into panel a and b for clarity.

Figure 4 and/or associated text: references should be provided for each strategy.

We have added references to the legend for each strategy.

Lines 177, 250 and following occurrence: specify that the evaluation of morbidity is by weight loss (not just loss).

We have clarified ‘weight loss’ in all occurrences.

Reviewer 3 Report

The review entitled “Strategies Targeting Hemagglutinin as a Universal Influenza Vaccine” by Brianna L. Bullard and Eric A Weaver is well written and articulated.

This is an excellent contribution to the influenza vaccine design. Although many efforts have been laid towards the development of universal influenza vaccines, very few are working against seasonal flu viruses. Therefore, this review would be a knowledge source for the researchers looking for multiple vaccine design strategies. The authors have comprehensively summarized the current knowledge about HA-based vaccine designs.

Here are some minor comments

VLP not expanded at first instance, in spite, expanded later multiple times.

Line 224: expand “VSV”

Line 230: “participants which” ----- “participants who”

Author Response

Dear Reviewers,

Thank you for your thoughtful suggestions. Please see the responses to your comments below. You will find that we corrected the manuscript and included the inserted text in the response in order to help you identify improvements. We thank the reviewers for their time and effort. 

Sincerely,

Eric Weaver

Reviewer 3

The review entitled “Strategies Targeting Hemagglutinin as a Universal Influenza Vaccine” by Brianna L. Bullard and Eric A Weaver is well written and articulated.

This is an excellent contribution to the influenza vaccine design. Although many efforts have been laid towards the development of universal influenza vaccines, very few are working against seasonal flu viruses. Therefore, this review would be a knowledge source for the researchers looking for multiple vaccine design strategies. The authors have comprehensively summarized the current knowledge about HA-based vaccine designs.

Here are some minor comments

VLP not expanded at first instance, in spite, expanded later multiple times.

We have defined virus-like particle (VLP) at the first occurrence.

Line 224: expand “VSV”

We have defined vesicular stomatitis virus (VSV).

Line 230: “participants which” ----- “participants who”

Corrected
